# Flying without a Net: Space Radiation Cancer Risk Predictions without a Gamma-ray Basis

**DOI:** 10.3390/ijms23084324

**Published:** 2022-04-13

**Authors:** Francis A. Cucinotta

**Affiliations:** Department of Health Physics and Diagnostic Sciences, University of Nevada Las Vegas, Las Vegas, NV 89154, USA; francis.cucinotta@unlv.edu

**Keywords:** high-LET carcinogenesis, radiation quality factors, space radiation, heavy ions, relative risk models, mars exploration

## Abstract

The biological effects of high linear energy transfer (LET) radiation show both a qualitative and quantitative difference when compared to low-LET radiation. However, models used to estimate risks ignore qualitative differences and involve extensive use of gamma-ray data, including low-LET radiation epidemiology, quality factors (QF), and dose and dose-rate effectiveness factors (DDREF). We consider a risk prediction that avoids gamma-ray data by formulating a track structure model of excess relative risk (ERR) with parameters estimated from animal studies using high-LET radiation. The ERR model is applied with U.S. population cancer data to predict lifetime risks to astronauts. Results for male liver and female breast cancer risk show that the ERR model agrees fairly well with estimates of a QF model on non-targeted effects (NTE) and is about 2-fold higher than the QF model that ignores NTE. For male or female lung cancer risk, the ERR model predicts about a 3-fold and more than 7-fold lower risk compared to the QF models with or without NTE, respectively. We suggest a relative risk approach coupled with improved models of tissue-specific cancers should be pursued to reduce uncertainties in space radiation risk projections. This approach would avoid low-LET uncertainties, while including qualitive effects specific to high-LET radiation.

## 1. Introduction

Space travel was preceded by human exposure to X-rays and gamma-rays, including for medical patients, in the atomic bomb detonations in Hiroshima and Nagasaki, Japan, and through the use of nuclear energy. This has led to risk assessments for high linear energy transfer (LET) radiation to be strongly coupled to low-LET epidemiology data. In this paper, I consider an alternative approach based on excess relative risk (ERR) factors derived by mouse tumor induction experiments with high-LET radiation (neutrons and heavy ions). Human studies with high-LET radiation are severely limited due to the population size of exposed groups, uncertainties in dosimetry, etc. Nuclear energy workers have largely received much lower doses of low-LET radiation than is of concern for space travel. Cancer epidemiology data are also not available for space radiation exposures because of the small number of humans that have traveled to space, while missions with a longer duration than those which have occurred in the past are the concern for the future, with missions outside the protection of the earth’s magnetic field being the main challenge for risk assessment. Relative risk models are the most common approach in medical studies on an unknown health challenge, so in essence, the present study considers a possible approach if the use of nuclear weapons had not occurred, while space travel proceeded.

Findings from human epidemiology studies on low-LET radiation (gamma-rays or X-rays), most frequently the Lifespan Study (LSS) of atomic bomb survivors, [1] have been the basis of predictions for cancer risks from space radiation [2,3], which are applied using a quality factor (QF) to weight organ doses for the purpose of extrapolating low-LET epidemiology data on space radiation exposure. Age-specific hazard rates are then formulated using background cancer rates in the population of interest, with this approach denoted as the conventional risk estimation model. Radiation epidemiology data have been invaluable in identifying the most radiogenic cancers (leukemia, breast, lung, etc.) and differences in risks associated with sex and the age at exposure. However, the use of epidemiology data in this manner leads to a variety of uncertainties including statistical, dosimetry errors, and differences in the population makeup and time periods compared to the exposure group of interest, as well as radiation quality and dose-rate effects. A large part of the uncertainty in QFs is due to the variability in gamma-ray responses for tumor induction or surrogate cancer endpoints [4]. Experimental studies of cancer induction by gamma-rays in mouse and rat strains are often inconsistent with some strains showing a weak or non-demonstrable dose response [5], which limits the strains and tissues that are available to estimate QFs. In contrast, high-LET radiation has been shown to lead to demonstrable dose responses in a large number of mouse and rat strains [5,6,7,8].

The qualitative differences in biological effects between high- and low-LET radiation are not considered in QF-based models. Indeed, in experimental studies, there are a large number of qualitative differences found between high- and low-LET related to molecular and DNA damage, cell signaling, and tissue responses to solid cancer risks, which I briefly discuss below. Such differences, combined with the absence of heavy ion epidemiology data, suggest the need for considering alternative approaches to space radiation risk assessments, especially for solid cancers.

The conventional model for cancer risk assessment is based on a hazard rate parameterized to epidemiology data, which are scaled to the background cancer rates in the population of interest, and the QF and a dose and dose-rate effectiveness factor (DDREF) estimate from gamma-rays. This leads to a model for the hazard rate with several parameters to be estimated for each tissue (T) at risk, which is applied to predict proton and heavy ion cancer risk (incidence (I) or mortality (M)) [4,9]:(1)$$\lambda_{IT}\left(a_{E},a,Z,E,D_{T}\right)=\lambda_{0IT}\left(a\right)\frac{\lambda_{\gamma IT,Epi}}{\lambda_{0IT,Epi}}\frac{\alpha \left(Z,E\right)}{\alpha_{\gamma }}\frac{D_{T}}{DDREF_{\gamma }}$$
where *a* is the attained age, *a_E_* age at exposure, *Z* and *E* the ion charge number and kinetic energy, and *D_T_* the tissue absorbed dose. The rates, *λ*, on the right-hand side of Equation (1) are the background age and tissue-specific cancer rate in the population where risks are to be estimated, and the background and gamma-ray cancer induction rates estimated from the population used in epidemiology studies. The middle quotient in Equation (1) is the relative biological effectiveness (RBE), which is the ratio of estimates of the linear dose response coefficients for the ions and gamma-rays, or alternatively for the QF. The denominator of the last quotient is the DDREF. The DDREF is estimated as a ratio of acute-to-low dose-rate exposure effects or from the amount of curvature in the acute gamma-ray response for cancer induction [1,9,10].

The main purpose of the present report is to consider an alternate model that is independent of gamma-ray data. The approach is based on direct application of the excess relative risk (ERR) model from experimental studies and is denoted as the direct excess relative risk (DERR) model. I use a retrospective analysis from small animal tumor induction studies; however, it should be expected that the prospective design of experiments would improve the application of this approach. In the DERR model, the hazard rate is estimated as:(2)$$\lambda_{IT}\left(a_{E},a,Z,E,D_{T}\right)=\lambda_{0IT}\left(a\right)\frac{\lambda_{IT,Expt}\left(Z,E\right)}{\lambda_{0IT,Expt}}D_{T}$$
where the ratio in Equation (2) is the ERR. The apparent advantage of Equation (2) over Equation (1) is the removal of uncertainties in gamma-ray effects from space radiation predictions. However, as such this relies on experimental findings with heavy ions and other high-LET radiation in a distinct manner from the conventional model. The model is decoupled from radiation epidemiology data, although observations from these studies could be combined in the present approach as described below. Equation (2) has to consider age at exposure or time since exposure (latency) in a manner distinct from the conventional model of Equation (1) due to this decoupling from epidemiology data. However, for adults of age 30 to 55 y, typical of the ages of astronauts, and for estimation of lifetime risks, it is likely not a large hurdle in the DERR approach. For example, the BEIR VII Report [1] ignored any age dependence of the hazard rate for exposures above age 30 y. A large part of the age dependence above age 30 y is due to the life table (declining years remaining with increasing age of exposure).

Table 1 compares sources of uncertainty in the conventional model to the DERR approach. Probability distribution functions (PDFs) describing the various uncertainties were described previously [1,3,11]. Dosimetry errors have been well studied in the LSS, leading to errors >15% [1], while retrospective organ dose assessments in other exposed populations often have larger errors. Extrapolation approaches in the conventional model include transfer model uncertainties, which are tissue dependent, with a larger uncertainty if differences between the A-bomb survivors and the population of interest’s background cancer rates occur. For a DERR model, this uncertainty is dependent on the quality of the mouse models developed through research, while previous mouse models have an uncertainty difficult to quantify. Dose estimation with particle accelerators carries small measurement errors (<5%).

## 2. Results

I fit the model of Equation (3) and a linear dose response model for the ERR for several data sets of mice or rats for hepatocellular, lung, and mammary carcinoma. I chose data sets (Table 2) when data for more than one animal strain (in most cases) was reported for high-LET radiation at doses < 0.5 Gy. I also only considered lifespan studies or studies with sufficient follow-up times that represent most of the animal’s lifespan. This approach avoids a parametrization influenced by a possible reduced latency at early time points for high-LET exposures. Because of restricted follow-up times and the slower progression of spontaneous tumors compared to radiation-induced tumors, ERR values may have a strong age dependence for limited follow-up times [12]. Of note is that the Harderian gland prevalence studies suggest ERR >10 occurs for Fe particles at 600 days prevalence [13,14]. However, the spontaneous tumor rate increases from ~2.9% at 600 days to ~10% at 800 days [12,15], and a declining ERR with time after exposure occurs, suggesting the use of an age-adjusted ERR [6]. The time dependence also influences RBE estimates; however, this is typically ignored. A similar observation is well known in experiments with Sprague-Dawley (SD) rats for induction of mammary tumors (benign or malignant), where a sufficient time shift to earlier appearances of high-LET radiation occurs as described in an NCRP Report [5].

Figure 1 shows results for hepatocellular carcinoma for several mouse strains. Results for heavy ions in CBA and C3H mice (Figure 1A,B) are similar, as shown by the parameter values listed in Table 3. The ERR saturation model led to improved fits compared to a linear response model. Experiments in B6CF1 male mice analyzed by Storer and Fry [7] considered low doses of acute and fractionated fission neutrons (mean energy ~2 MeV); however, because the results were not significantly different, I pooled the data in our fits. Here, the linear model fits better (Figure 1C). However, the lack of higher doses in the study does not allow for a study of a bending in the dose response or for a possible saturation of the response at higher doses. Figure 1D shows the results for ions in CBA and C3H mice and for fission neutrons of lower mean neutron energy (0.45 MeV) by Di Majo et al. [8]. For these comparisons, only the neutron dose estimate is used. Figure 1D includes the application of the ERR model of Equation (3) using Equation (7) for Si, Fe and fission neutrons. The Fe particles are more efficient per unit fluence (per particle); however, at identical doses, the fluence of recoil particles from fission neutrons or Si ions is about 2-fold higher than Fe.

Dose response data for induction of lung adenocarcinoma were reported for female Balb/C and B6CF1 mice: acute and chronic adenocarcinoma in Balb/C and fractionated adenocarcinoma in B6CF1 mice with fission neutron exposures (Figure 2A,B). The saturation ERR model provided a better fit to the data for Balb/c mice, while the linear model was a better fit to the fractionated B6CF1 mice data. Figure 2C shows results for fractionated exposures in female and male B6CF1 mice. For lung carcinomas in mice, there are data for cyclotron neutrons in male and female SAS/4 mice [16] (Figure 2D); however, these data present the complication of inelastic scattering by neutrons and the use of thorax-only irradiation. Cyclotron neutrons will also have a lower effective LET compared to fission neutrons or Si and Fe particles, which limits direct comparisons to the heavy ion results. The use of partial body irradiation leads to a higher RR compared to the other experiments. For both the B6CF1 and SAS/4 strains, female mice have higher ERR values compared to male mice. For lung cancer in male mice, we considered the B6CF1 mouse dose fractionation data with fission neutrons [7]. A regression fit to these data did not converge, while a linear response provided a very good fit (Table 2 and Figure 3).

**Table 2 ijms-23-04324-t002:** Sources of tumor data in mice and rats exposed to fission neutrons (FN) or heavy ions. Data are chosen as representative of major cancer types in humans where more than one animal strain was reported at doses of high-LET radiation below 0.5 Gy, and with observation periods for tumor appearance representative of the animal lifespan. Doses used in the various studies are shown in the Figures of the present report.

Model	Tumor	Age at Irradiation	Irradiation Types	Duration of Observations	Reference
BALB/c/AnNBdf, female mice	Mammary adenocarcinoma	120 d	FN mean energy 2 MeV, acute and chronic	Lifespan	[6]
BALB/c/AnNBdf, female mice	Lung adenocarcinoma	120 d	FN mean energy 2 MeV, acute and chronic	Lifespan	[6]
CBA/CaJ, male mice	Hepatocellular carcinoma	8–14 weeks	^56^Fe (1 GeV/u), acute	Observed to age 800 d	[17]
C3H/HeNCrl, male mice	Hepatocellular carcinoma	8–10 weeks	^56^Fe (0.6 GeV/u), ^28^Si (0.3 GeV/u), acute	Observed to age 800 d	[18]
Sprague-Dawley, female rats	Mammary carcinoma	7 weeks	Fast neutrons mean energy 2.3 MeV, acute	Observed to age 90 weeks	[19]
B6CF1 ((C57BL/6 Bd x BALB/c Bd), male mice)	Hepatocellular carcinoma	16 weeks	FN mean energy 2 MeV, acute and chronic	Observed to 1200 d	[7]
B6CF1 (C57BL/6 Bd x BALB/c Bd), female and male Mice)	Lung Adenocarcinoma	16 weeks	FN mean energy 2 MeV, acute and chronic	Observed to 1200 d	[7]
BCF1, male mice	Hepatocellular carcinoma, all solid cancers	3 months	FN mean energy 0.4 MeV, acute and fractionated	Lifespan	[8]

**Figure 1 ijms-23-04324-f001:**
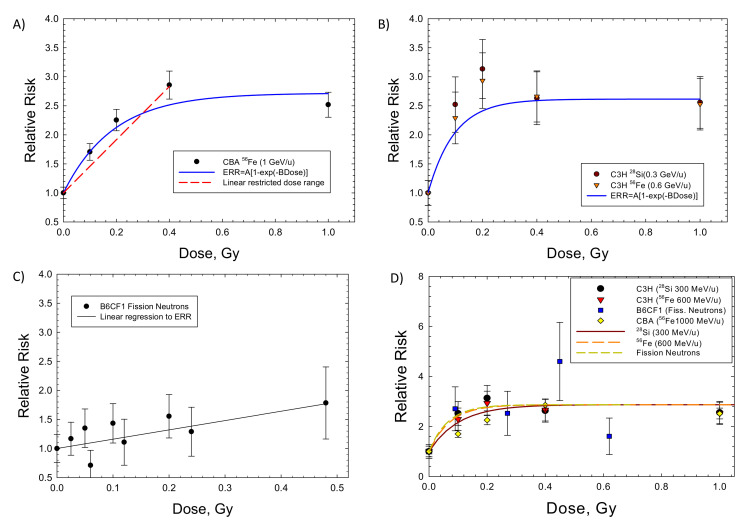
Dose responses for the relative risk of hepatocellular carcinomas in several strains of male mice exposed to heavy ions (HIs) or fission neutrons (FN). The saturable ERR model described in the text is compared to the experimental data. (**A**) CBA exposed to HIs [17], (**B**) C3H exposed to HIs [18], (**C**) B6CF1 exposed to FN [7]. (**D**) Model fits to several strains of mice showing similarity of HI and FN responses from [8].

**Table 3 ijms-23-04324-t003:** Parameter estimations for the ERR function for several strains of mice or rats: lung and mammary tumors for females and liver tumors for males.

	Saturation Model				Linear Model	
Tissue: Radiation	A	B, Gy^−1^	AB, Gy^−1^	Adjusted R^2^	α, Linear, Gy^−1^	Adjusted R^2^
**Lung Balb/c (F): FN** **(chronic)**	1.43 ± 0.365 (*p* < 0.0175)	6.01 ± 3.29 (*p* < 0.1415)	8.59	0.893	4.12 ± 0.47 (*p* < 0.003)	0.883
**Lung B6CF1 (F): FN** **(fractions)**	1.24 ± 1.69 (*p* < 0.518)	2.99 ± 7.04 (*p* < 0.7)	3.71	0.585	3.25 ± 0.34 (*p* < 0.0007)	0.918
**Lung B6CF1 (M):** **FN** **(fractions)**	-	-	-	NC	1.83 ± 0.115 (*p* < 0.0001)	0.966
**Mammary Balb/c: FN (chronic)**	2.29 ± 0.068 (*p* < 0.0001)	29.77 ± 3.36 (*p* < 0.0009)	68.2	0.986	4.35 ± 2.85 (*p* < 0.19)	−1.17
**Mammary Balb/c: FN (acute)**	1.95 ± 0.258 (*p* < 0.0016)	9.21 ± 2.75 (*p* < 0.029)	17.96	0.9491	5.05 ± 1.21 (*p* < 0.0096)	0.557
**Mammary SD Rats:** **2 MeV neutrons** **(acute)**	1.81 ± 0.326 (*p* < 0.0052)	4.99 ± 2.19 (*p* < 0.0857)	9.03	0.867	1.804 ± 0.596 (*p* < 0.092)	0.328
**Liver C3H (M): Si (0.3 GeV/u) and Fe (0.6 GeV/u) (acute)**	1.61 ± 0.2 (*p* < 0.0001)	10.28 ± 4.9 (*p* < 0.0691)	16.6	0.799	0.45 ± 0.96 (*p* < 0.652)	−5.14
**Liver CBA (M):** **Fe (1 GeV/u) (acute)**	1.72 ± 0.22 (*p* < 0.0044)	5.33 ± 1.79 (*p* < 0.058)	9.2	0.937	4.78 ± 0.53 * (*p* < 0.0032)	0.906
**Liver B6CF1 (M): FN** **(acute and fractions)**	1.749 ± 0.362 (*p* < 0.0019)	26.02 ± 24.96 (*p* < 0.332)	45.5	−0.23	1.603 ± 0.365 (*p* < 0.0023)	0.521
**All solid B6CF1 (M): FN (acute and fractions)**	1.69 ± 0.39 (*p* < 0.0034)	4.64 ± 2.4 (*p* < 0.097)	7.84	0.825	2.69 ± 0.5 (*p* < 0.006)	0.503

Model functions are the saturation model, ERR = A [1 − exp(−B Dose)] or linear model, ERR = α Dose. Means, standard deviations and adjusted R^2^ values are listed. Abbreviations NC = Regression did not converge, SD = Sprague-Dawley, FN = fission neutrons. * Restricted dose range is assumed as shown in Figure 1.

For mammary tumors in mice, the results for Balb/C mice are the only available dose response data for high-LET radiation. Storer and Fry [7] note that the B6CF1 strain is not susceptible to radiation-induced mammary tumors. I considered the recent data of Imaoka et al. [19] using 7-week-old rats exposed to a neutron spectrum with a mean energy of 2 MeV because of the long follow-up period of 70 weeks, approaching a lifespan study. Figure 4 and Table 3 show that the saturable ERR model fits the acute data in Balb/c mice and SD mice with similar saturation values but with a stronger response in Balb/c. An ERR enhancement is observed for the chronic exposures in Balb/c mice [5], with a higher saturation value of 2.29 compared to 1.95 in the acute exposures with fission neutrons. Fits to the various mammary tumor data with a linear ERR model did not lead to a significant result. A similar excess incidence occurred for combined carcinomas and fibromas. However, data on the spontaneous tumor rates were not reported and would be needed to make a numerical estimate of ERR. Di Majo et al. [8] also reported dose response data for all solid cancers. Dose fractionation using five daily doses were used in the experiments. Figure 5 shows that the saturable ERR model provides a good fit to these data. Data for all solid cancers were not reported for B6CF1 mice by Storer and Fry [7]. 

**Figure 2 ijms-23-04324-f002:**
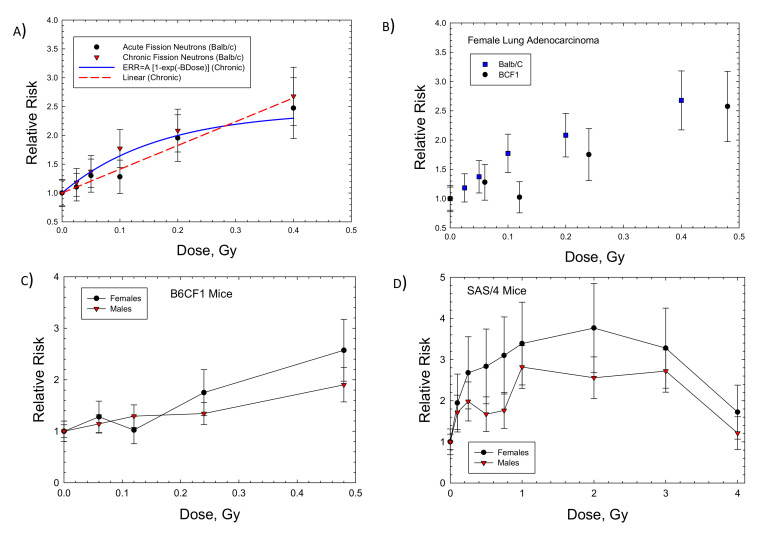
Dose responses for relative risk of lung adenocarcinomas in female Balb/c, B6CF1, or SAS/4 mice exposed to neutrons. (**A**) Female Balb/c mice exposed to fission neutrons (FN) showing linear and saturable ERR models [6]. (**B**) Comparison of Balb/c to B6CF1 mice exposed to FN. (**C**) Comparison of female to male B6CF1 mice exposed to FN. (**D**) Comparison of female to male SAS/4 mice exposed (thorax only) to cyclotron neutrons [16].

**Figure 3 ijms-23-04324-f003:**
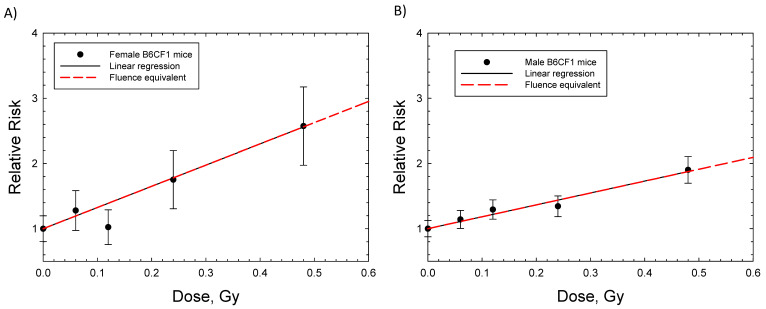
Linear dose responses for relative risk of lung adenocarcinomas in female and male B6CF1 mice exposed to daily fractions of fission neutrons [7]. (**A**) Female B6CF1mice, (**B**) Male B6CF1 mice. Symbols are experimental data with standard errors. Solid (black) line is linear regression and dashed red line is the fluence base result after integrating over charged particles produced by neutrons.

**Figure 4 ijms-23-04324-f004:**
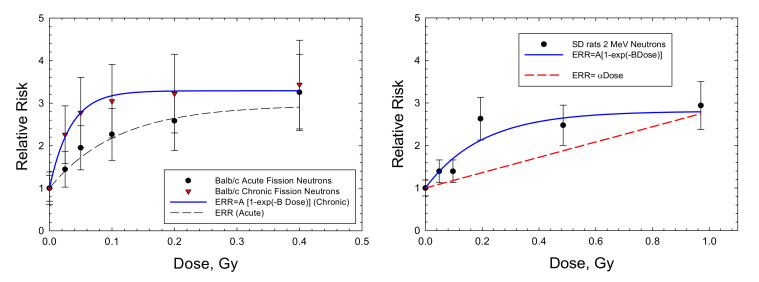
Dose responses for relative risk of mammary adenocarcinomas in Balb/c mice [6] and Sprague-Dawley rats [19] exposed to fission neutrons or neutron sources with average energy 2 MeV. The saturable ERR model described in the text is compared to the experimental data.

**Figure 5 ijms-23-04324-f005:**
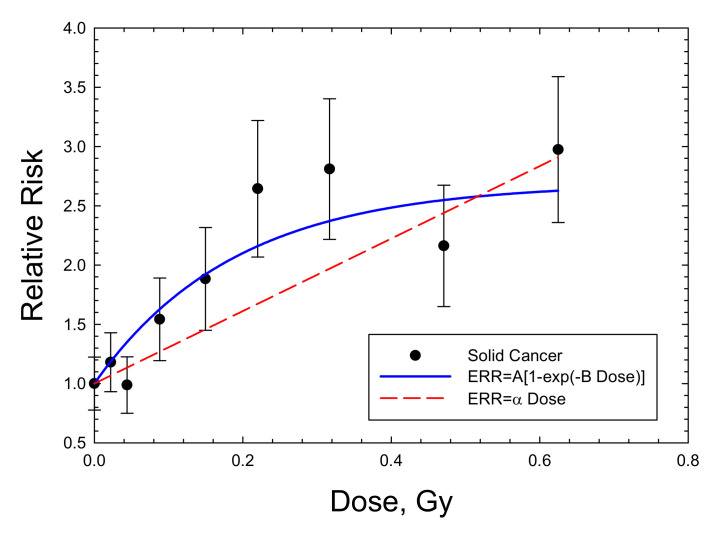
Dose response for relative risk of solid cancers in male B6CF1 mice exposed to fission neutrons (mean energy 0.4 MeV). The saturable ERR model described in the text is compared to the experimental data of [8].

I considered the parameters of the various fits in Table 3 that allow us to make predictions of cancer risks for GCR and compare them to the NSCR-2020 model. The preferred parameter values are shown in Table 4. For the uncertainty in the A_T_ and Σ_T_ parameters, I assumed normal distributions with a standard deviation of 33% of the mean. For mammary cancers, the chronic exposures used by Ullrich [6] are preferred for risk assessment for astronauts; however, the parameter values were adjusted to reflect the average of the acute exposures in Balb/c mice and SD rats. The analysis of Storer and Fry [7] for lung tumors in B6CF1 mice exposed to daily fractions of neutrons is restricted to lower doses, which precludes the observation of a likely saturation effect. In contrast, for lung cancer in Balb/c female mice chronically exposed to fission neutrons, the saturation model provided a better fit. Therefore, I focus on the saturable ERR model for space radiation calculations as described below for mammary and liver tumors, while considering both the linear and saturation models for lung cancer. Estimates of the uncertainties in other DERR model parameters are limited at this time because very few experiments are available, and therefore I use values in the NSCR-2020 model for several of the parameters.

Predictions for annual GCR exposure for missions at average solar minimum conditions are shown in Table 5 for 35-y- and 50-y-old astronauts. Predictions include those for breast, liver and lung cancers in females and liver and lung cancers in males. The NSCR-2020 [20] model is used to make predictions with and without an estimate of NTEs on the QF. Interestingly, the results for breast and liver cancers with NTEs agree closely to those of the DERR model. The age dependence is similar since the largest contributor to declining risk above age 30 y and with an increasing age of exposure is the constraint of the life table, which is identical in the models presented using U.S. population data. Predictions for lung cancer risks in the DERR model for both females and males suggest about a 3-fold and >7-fold lower risk compared to NSCR-2020 with and without NTE, respectively.

Predictions of cancer risks for the U.S. population for females are higher than males for several types of cancer including stomach, liver, urinary track, and all solid cancers. However, the largest difference is for lung cancer where the F:M incidence ratio is 2.83 [21]. It is not clear if this difference is reflected in mouse experiments. A higher ERR for females compared to males occurs for lung cancer in thorax-only irradiation of cyclotron neutrons in SAS mice [16] and whole-body exposures to FNs in B6CF1 mouse [7].

## 3. Discussion

Space travel began after the atomic bomb exposures in Japan, the development of nuclear energy, and the use of radiation in cancer therapy or diagnostically. This has led to the reliance of cancer risk estimates for cosmic ray exposures using an approach based on the application of epidemiology data for low-LET radiation exposures [3,4,22,23,24]. However, radiobiology studies have shown that a large number of qualitative differences occur between low- and high-LET radiation, including in the initial molecular damage, cell signaling, and tissue effects. Therefore, an important question remains as to how to treat the possible qualitative differences not described by QFs.

In the area of DNA damage and repair, due to the high density of ionizations, high-LET radiation produces much larger amounts and unique types of clustered DNA damage [25], leading to distinct cytogenetic changes [26,27], damage repair choices, and other damage responses [28,29]. For example, an enzyme-stimulated chromosomal rearrangement was used as an endpoint by Li et al. [30] in a study of lung epithelial cell cultures that were irradiated with energetic ^48^Ti ions or gamma-rays. After a six-day recovery, cells were challenged by expression of a Cas9/sgRNA pair that creates DSBs simultaneously in the EML4 and ALK loci leading to the EML4–ALK fusion oncogene. In contrast to the controls or gamma-rays, this delayed mis-rejoining effect indicating genomic instability was significantly elevated in ^48^Ti-irradiated populations and were far more likely to contain deletions, sometimes flanked by short microhomologies. 

Low doses of high-energy heavy ions produce long columns of heavily damaged cells, while low doses of gamma-rays lead to a sparse pattern of damaged cells. This is a qualitative difference that is distinct from the molecular and DNA damage noted above. Experiments in tissues have demonstrated radiation quality differences in the biochemical signaling, tumor spectra and degree of malignancy in rodent models, inflammatory responses and genomic profiles [9,18,29,31,32,33,34]. In mouse lungs, McConnell et al. [34] showed that progenitor cells residing in distal airways are more radiosensitive than those in proximal airways, while high-LET radiation activates the tumor suppressor gene p53 in a distinct manner from low-LET radiation, leading to progenitor cells in the distal airway epithelium that undergo apoptosis and senescence, and create defects in mitotic progression. This decreases the pool of progenitor cells able to successfully proliferate and forces the remaining proliferative-competent progenitors to clonally expand in order to maintain homeostatic levels of turnover. Such information suggests that models that consider only quantitative differences have important limitations and uncertainties in predicting heavy ion risks.

The potential for qualitative differences between high- and low-LET radiation is apparent in murine studies of solid cancers [9,18,28,30], while mechanisms for leukemia induction seem to appear as quantitative differences alone for leukemia [35]. RBE values for leukemia and lymphomas in mice are modest (<5) for both heavy ions [9,17,18] and fission neutrons [5,6,36]. The latency and time-after-exposure dependency is distinct when comparing leukemia to solid cancer, which requires a more complex approach than a simple constant time-after-exposure ERR as used in this report. Such differences could be considered in future work for treating leukemia risk. 

In medical sciences, RR models are used extensively to assay risks or the effectiveness of potential treatments. For example, epidemiology data for low doses of gamma-rays are extremely sparse, and animal studies suggest cognitive risks will occur for low doses of heavy ions [37]. A model based on ERR was used to assay cognitive risks from GCR in a recent report, and in that report, ERR values for diseases such as Alzheimer’s and Parkinson’s diseases were discussed as comparisons [37]. In this paper, I used estimates from the excess relative risk model for high-LET rodent studies to predict cancer risks, and thus entirely bypassed low-LET data (epidemiology or radiobiology). The similarities along with differences of the DERR and NSCR-2020 models for central estimates of risk are of interest, as they suggest agreement of at least several fold, while the differences found suggest the need to include NTEs in risk models. Comparisons of the uncertainties in the models are limited due to the scarcity of data to estimate DERR model parameters, and therefore I relied on similar PDF estimates as in the NSCR-2020 model for the *m* and κ parameters.

Approaches to extrapolate from experimental data to humans have been limited by the conventional approach that has used QFs as the mainstay of space radiation risk predictions for several decades [2,3,11,22,23]. The National Council on Radiation Protection and Measurements (NCRP) considered extrapolation approaches for space radiation in the past, including fluence-based risk coefficients and microdosimetric approaches [38]. However, these approaches relied on gamma-ray data for predictions. Relative risk models were considered in a NCRP Report [39] that considered extrapolation from nonhuman systems to humans. Carnes et al. developed an approach to extrapolate mortality after radiation exposure between animal models and humans focusing on age-specific survival with a linear dose response model [39,40]. A Cox proportional hazard model describing underlying intrinsic and extrinsic causes of mortality was successful in predicting risks in several species comparing gamma-rays to neutrons in animal studies. However, this was performed with no consideration of the specific aspects of tissue-specific risks, the ion track structure or the deviation from a linear response for high-LET radiation.

In this paper, I considered direct application of ERR estimates from high-LET tumor induction studies to risk predations independent of gamma-ray data in their totality. Therefore, possible limitations in risk estimates that use gamma-rays as a basis, such as qualitative differences between high- and low-LET, are avoided. In the future, a model that combines the DERR approach with results from low-LET epidemiology could be developed as a hybrid approach, perhaps using a Bayesian method. 

An interesting result found is that lung cancer risk estimates in the DERR model are much lower than the conventional risk assessment used in NSCR. The use of tobacco products is a large confounder in lung cancer risk estimates [1,21]. Results from the LSS suggest a synergistic effect between radiation exposure and modest tobacco usage (<1 pack per day), with a diminishing radiation effect on risk for heavy smokers. Astronauts in the U.S. are largely lifetime never-smokers [41], and it is therefore useful to pursue estimates without such interactions. The ERR and risk estimates comparing females to male lung cancer risks in the DERR model suggest females have about a 2-fold higher risk than males, which is similar to that found in the conventional risk model based on the LSS.

Because of the scarcity of heavy ion tumor dose response data, the variation in responses with ions *Z* and *E* are assumed to follow the same variation as observed in other radiobiology experiments [9,42], but calibrated to tumor induction studies through the A_T_ and Σ_T_ parameters. The mathematical forms are motivated by amorphous track structure approaches [43], but with similar radiation quality dependences found using frequency distributions in nanoscale volumes (<25 nm) based on Monte-Carlo track structure simulations [11,44]. Experiments with ions of *Z* and *E* values were the probability function of Equation (6) such that *P(E,Z)~*0.5 will have the maximum effectiveness per unit dose, and this condition approximately holds for fission neutrons and the energies of Fe and Si ions considered. Of note is that the values of Σ_T_ for breast and liver cancers, 1000 μm^2^ and 400 μm^2^, respectively, are much larger than the cross-sectional area of a cell nucleus, which is suggestive of a role for NTEs [31,32]. The differences found in the conventional and DERR model predictions for lung cancer are interesting due to the role of primary and secondary exposures to tobacco products as a confounder of epidemiology data, suggesting an important area of future research.

The use of age-adjusted experimental data is a potential limitation of the present estimates. However, I note that not only ERR functions but also RBEs likely have time-after-exposure, latency or age dependencies [12]. The collection of detailed time-after-exposure tumor data requires larger animal studies than most studies in the past and also introduces new questions for extrapolation models with regard to the translation of portions of an animal lifespan to that of humans. Age-adjusted values avoid these complications, but introduce uncertainties for both the conventional (QF) and DERR models that are not readily quantified.

Data for mammary cancers in mice are limited to the Balb/c strain exposed to neutrons as noted above. However, mechanistic studies on mammary tumor induction with heavy ions point to qualitative differences with gamma-ray exposures [31,33]. There are several data sets for SD rats with neutrons or heavy ions [45,46,47,48]. I considered the Imaoka et al. [19] study of 7-week-old rats exposed to a neutron spectrum with a mean energy of 2 MeV because of the long follow-up period of 70 weeks, approaching a lifespan study. Figure 4 and Table 3 show that the saturable ERR model fits the acute data for Balb/c mice and SD mice with a similar saturation value. A similar saturation of responses in SD rats is seen by Dicello et al. [47], with Fe particle exposures reporting excess incidence of combined mammary carcinomas and fibromas. However, data on the spontaneous tumor rates were not reported and would be needed to make a numerical estimate of ERR.

The uncertainties in physics estimates are a minor contribution to the overall uncertainty. The major contributor to the uncertainty estimates in the DERR model is the κ and Σ_T_ parameters. This is similar to the conventional form of the NSCR model with κ and a parameters that normalizes and effectively cross sections the main contributors, but with additional uncertainties in the low-LET data inputs of the model. The uncertainty in the values of the κ and Σ_T_ parameters could be substantially reduced with a prospective experimental design, rather than the conservative estimates made here based on the scarcity of historical data. Beyond these uncertainties, there is the uncertainty of the model systems used to estimate either a QF and DDREF or the ERR in the two approaches. This type of uncertainty pervades both approaches.

A main focus of this report is that experimental models that are designed to accurately represent tissue-specific cancer risks should play the major role in reducing uncertainties in risk projection, and are also needed for realistic countermeasure studies. The parameter values obtained suggest the heterogeneity of responses in the small number of rodent models considered. Prior reviews have suggested a number of design criteria for experimental systems to represent human risks, which include a focus on the tissues which represent the largest contributors to risk, which are lung, breast, colon, stomach, liver and leukemias for adults (>30 y). In addition, the cells of origin for cancer and a genetic diversity similar to humans should be considered. The study of radiation cancer susceptibility in mice has been shown to reflect spontaneous rates of cancer development [48,49]. Genetic factors related to cancer have strong overlap between humans and mice [50], but differences are known to occur [51,52]. A practical limitation of the ERR approach would be its application to models, such as for transgenic mice, that have small or negligible frequency of spontaneous cancers. However, such models are not likely to represent human risk, albeit mechanistic hypotheses are effectively investigated with such approaches [53]. The use of genetically diverse cohorts [54] are a possible approach to this problem, although the number of dose groups (~5 to 6) needed to accurately define a dose response for several ions is likely prohibitive due to the large number of mice used in these studies.

## 4. Materials and Methods

We fitted data using Sigmaplot (version 12.5) for the relative risk (RR) of the dose response from heavy-ion- or fission-neutron-induced lung, mammary, and hepatocellular tumors in mice or rats. We consider age-adjusted data or data for tumor induction over the lifespan of a mouse [55,56]. This approach avoids any time dependence in the RR or ERR estimates. The excess relative risk is ERR = RR − 1, and we considered both a linear function and the function
(3)$${\mathrm{EER}}_{T}=A_{T}\left[1-\exp\left(-B_{T}D_{T}\right)\right]$$

This function describes the bending over and saturation of high-LET dose responses observed in the experiments. The parameter *A_T_* is seen to represent the saturation value of the ERR, while at a sufficiently low dose, the ERR becomes linear in dose with the coefficient *A_T_ B_T_*. In contrast to Equation (3), a linear response model simply assumes ERR*_T_ = constant Dose.* To predict the risk of exposure-induced cancer (REIC), the ERR function is folded with tissue-, age-, and sex-specific spontaneous cancer rates and the probability to survive to a given age. This probability is then integrated from the age of exposure to the remainder of a life (taken as 100 years) as described previously [9,11]. Cancer incidence, mortality, and life table data for the U.S. average population are used in the calculations. The same cancer and life table data are used for the conventional model and the DERR model, with the ERR values from the LSS [21,57,58] used in the conventional model. 

Equation (3) is then recast as the ERR in terms of the tissue-specific ion fluence, F_T_, for charge number Z and kinetic energy Z as:ERR(Z,E,F_T_) = A_T_ [1 − exp(−Σ_T_F_T_)](4)

The normalization parameter, Σ_0T_ (in units of area, μm^2^), of the effective action cross section is determined by B_T_ considering the details of the energy spectra in the mouse tumor induction experiments with neutrons or heavy ions. The effective action cross section, Σ_T_, in Equation (4) is represented by the form:Σ_T_(E,Z) = Σ_0T_P(E,Z)(5)

We use a mathematical form motivated by track structure models in the NSCR model:P(E,Z) = [1−exp(−Z/κβ^2^)]^m^ [1−exp(−E/0.2)](6)
where m and κ are model parameters, β is the ion velocity scaled to the speed of light and *Z** is the effective charge number that describes the electron pickup as the ion’s velocity decreases. The function P(E,Z) allows us to extrapolate from the ions or neutrons considered in experiments to other ions of a given Z and E based on a large number of prior studies of mice and cell culture models [9,42] to estimate values of the parameters κ and *m*.

We use the above ERR descriptions to predict the risk of exposure-induced cancer (REIC) for annual GCR exposures, and performed uncertainty analysis using Monte-Carlo methods as described previously [9,11]. The *λ*_I_ (or *λ*_M_) is a sum over rates for each tissue that contributes to cancer risk, *λ*_IT_ (or *λ*_MT_). The REIC is calculated by folding the instantaneous radiation cancer incidence rate with the probability of surviving to time *t,* which is given by the survival function *S*_0_*(t)* for the background population times the probability for radiation cancer death at a previous time, summed over a space mission exposure and then integrated over the remainder of a lifetime.

Lifetables from the U.S. Center for Disease Control and Prevention (CDC) for male and females in the U.S. are used in the calculations [59]. Tissue-specific cancer incidence and mortality rates are available from the National Cancer Institute’s Surveillance Epidemiology and End Results (SEER) program, with the latest data collected for the years 2014–2018, which provides race-, ethnicity-, age-, and sex-specific rates. For cancer incidences, we used the SEER delay-adjusted rates, which estimate the impact of delays in reporting of cancer cases [60,61].

### Fission Neutron and Space Radiation Organ Exposures

To apply the models described above, the charged particle energy spectra are needed. Laboratory-based heavy ion experiments are assumed to be in the track segment model, neglecting nuclear fragmentation in the beam line [62]. Fission neutrons have energies largely below 5 MeV, which simplifies the calculation of particle spectra for the elastic recoils produced in tissues (H, C, O) and gamma-rays. Based on earlier findings of Storer et al. [63], the experiments of Ullrich [6] and Storer and Fry [7] ignore gamma-ray contributions to absorbed doses due to the much larger biological effectiveness of neutrons with energy <5 MeV. We followed this approach for the various experiments considered using neutrons. Charged particle spectra for a ^252^Cf fission source developed by Dennis and Edwards [64] are used for calculations, which are folded with the model functions to fit the parameters to the experimental data.

GCR exposures include primary and secondary H, He, and HZE particles, and secondary neutrons, mesons, electrons, and γ-rays over a wide energy range. We used the HZE particle transport computer code (HZETRN) with the quantum fragmentation model nuclear interaction cross sections [65,66,67] to estimate the particle energy spectra for particle type *j*, *φ_j_(Z,E)* as described previously by Cucinotta et al. [9]. GCR organ dose equivalents show little variation from 10 to 50 g/cm^2^ of shielding, and we use 20 g/cm^2^ for calculations, which is a typical average shielding amount [11].

A mixed-field action cross section is formed by weighting the particle flux spectra, *φ_j_(E*), for particle species, *j*, contributing to GCR exposure evaluated with the HZETRN code with the pseudo-biological action cross section for mono-energetic particles and summing over all particles and kinetic energies [9]: (7)$$(\Sigma F{)}_{T}=\Sigma j\int \varphi j\left(Z,E\right)\Sigma \left(Z,E\right)dE$$

## 5. Conclusions

The main conclusion of this report is that space radiation cancer risk predictions independent of the low-LET epidemiology, such as with the LSS, the QF, and DDREF, are readily performed for several tissues with existing high-LET tumor data, and the results are similar to the NTE estimates of the NSCR model, with the exception of lung cancer in females. The DERR model includes possible qualitative differences not found in low-LET radiation effects, which are not described by QFs. However, the present results are limited due to the models reported in the literature, and I suggest a large uncertainty reduction could be obtained by developing experimental models of the major tumors at risk from radiation in humans. The uncertainty estimates made herein are similar to the NSCR model; however, this is largely due to the use of identical values for the *m* and κ parameters and the associated PDFs of uncertainty in the model, which are applied here due to the scarcity of data necessary to apply the DERR approach. In future work, I will extend the analysis to data from higher energy neutrons and consider time-dependent models of ERR as a function of radiation quality using more detailed models of neutron transport than considered in the present report. The use of time-after-exposure-dependent ERR models will allow for parametrizations of experiments limited by follow-up time, while the present report considered only lifespan studies. In addition, an approach to include information from low-LET studies in the action cross section model and other aspects of the risk DERR models will be considered.

## Figures and Tables

**Table 1 ijms-23-04324-t001:** Comparisons of sources of uncertainty in the conventional model to the DERR model.

Source of Uncertainty	Conventional Model	DERR Model
Statistical	Epidemiology data, experimental for gamma-rays, experimental for ions.	Experimental for ions alone.
Dosimetry	Epidemiology data, and in accelerator studies with animals.	In accelerator studies with animals.
Time period	Epidemiology data, including changes in background cancer rates over decades due to changes in environmental, dietary, and genetic factors.	Small if models developed in reasonable research time period.
Genetic and environmental factors	Likely distinct in epidemiology data from modern day radiation workers including astronauts. Considered in experimental design for gamma-rays and ions.	Considered in experimental design for ions.
Dose and Dose-rate	Extrapolation of epidemiology data, and experimental data for gamma-rays to risks of ions at chronic low doses. Uncertainty in shape of response at low dose.	Extrapolation of experimental data for ions to chronic low doses.
Range of doses and ion types considered	Limited by investments in experiments performed.	Limited by investments in experiments performed.
Extrapolation of tissue-specific factors from epidemiology or mouse studies to humans.	For both gamma-rays and ions. Transfer model uncertainties.	For ions.
Space dosimetry	Modest uncertainties in organ dose equivalent (<15%). Dosimetric methods developed.	Modest uncertainties in organ dose equivalent (<15%), but dosimetric verification methods have not been developed.

**Table 4 ijms-23-04324-t004:** DERR model preferred parameters for cancer risk in several tissues.

Tissue	A_T_	B_T_, Gy^−1^ (Σ_0_, μm^2^)
Lung (Females)	1.4 ± 0.4	5 ± 2 (250)
Liver (Males)	1.65 ± 0.2	7.5 ± 3 (400)
Breast (Females)	2.1 ± 0.3	20 ± 5 (1000)
	**Σ_T_, μm^−1^**	
Lung (Females)	148	
Lung (Males)	83	

**Table 5 ijms-23-04324-t005:** Predictions of risk of exposure-induced cancer (REIC) for 1-year exposures with average solar minimum conditions for astronauts of 35 and 50 y at mission launch. Results for the NSCR-2020 model with and without non-target effects (NTE) are shown in comparison to the DERR model. Uncertainties for physics and the *κ* and *m* parameters are the same in each prediction as described previously [9,20]. Uncertainties in other parameters are distinct for the various predictive models.

Cancer Site	NSCR-2020	NSCR-2020 with NTE	DERR Saturation Model	DERR Linear Model
	**Age at Exposure = 35 years**			
**Lung Females**	1.98 [0.52, 6.0]	4.7 [1.4, 10.7]	0.92 [0.2, 3.1]	0.42 [0.07,1.43]
**Lung Males**	0.70 [0.18, 2.0]	1.64 [0.49, 3.84]	-	0.24 [0.04,0.83]
**Breast Females**	4.2 [0.81, 13.2]	9.7 [2.2, 23.5]	8.5 [2.1, 21.5]	-
**Liver Males**	0.11 [0.03, 0.34]	0.27 [0.08,0.63]	0.24 [0.05, 0.61]	-
	**Age at Exposure = 50 years**			
**Lung Females**	1.99 [0.51, 6.03]	4.68 [1.4, 10.7]	0.88 [0.17, 2.89]	0.41 [0.02,0.23]
**Lung Males**	0.73 [0.19, 2.24]	1.57 [0.48, 3.67]	-	0.24 [0.04, 0.86]
**Breast Females**	2.98 [0.57, 9.41]	7.0 [1.53,17.2]	6.8 [ 1.67, 16.2]	-
**Liver Males**	0.07 [0.02, 0.22]	0.17 [0.05, 0.40]	0.2 [0.05, 0.44]	-

## Data Availability

Not applicable.

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
