# Peer review of "Flying without a Net: Space Radiation Cancer Risk Predictions without a Gamma-ray Basis"

_ijms, 2022, doi:10.3390/ijms23084324_

Round 1

Reviewer 1 Report

This paper tried to estimate risk prediction with a model of ERR based on the data obtained from animal studies with high LET radiation that avoids low LET (DERR model), which is uncertain at low dose range, to predict lifetime risks to astronauts. Interestingly, the data for male liver cancer and female breast cancer risk indicated the similar estimates between a QF model with NTE (NSCR-2020 with NTE) and DERR model. In contrast, the data for female lung cancer risk were 5-fold differences between DERR model and NSCR-2020 with NTE. The model fitting approach is interesting and might be relevant to exposures to high LET radiation in the space, but there are a number of major concerns about the interpretation of data that the authors need to address:

Specific comments are as follows.

Critique 

Major points:

  1. Historically, a lot of studies with tremendous efforts have been done by scientists, particularly epidemiologists using the data of low LET based on atomic bomb survivors and nuclear accidents. Therefore, it is important to cite original important studies (e.g. Grant et al, Radiat Res, 2017, 187(5), 513-537) in parallel with the reports provided by the NCRP and BEIR to acknowledge their works.

  1. For readers who are not familiar with the field of radiation effects, it is hard to find the sentence the author wants to indicate in the references because NCRP reports consist with hundreds of pages and is time consuming to search. Therefore, it would be helpful if the author provides the precise information on the sentence in the NCRP reports (e.g. NCRP Report 132, page xxx, line xxx). Moreover, the readers may wonder whether report published in 1989 [3], 1990 [5] is updated or not.

  1. I understand that the biological effect data on the high LET in rodents is limited. However, the modelling attempt to fit the DERR model to the NSCR-2020 with NTE for assessing high LET is not sound to me, as one of the three cases (female lung cancer, male liver cancer, female breast cancer) failed to fit the model perfectly. Therefore, it would be interesting to examine another cancer case in rodents. I think this is an essential point, especially when the author wants to ignore the effect of low LET to substantiate the new risk model to assess the risk of high LET for space flight.

  1. I am not convinced that the author discussed the reason for the difference between DERR model and NSCR-2020 with NTE in female lung cancer as the role of primary and secondary exposures to tabaco products. Globally, smoking rates among females are lower than those among males in human (Ng et al, JAMA, 2014, 311, 183-92). Therefore, it is important to check the risk of lung cancer in male mice using the data Dr. Fry provided who assessed the risks of cancers induced by neutron in thousands of male mice ([7] Storer and Fry, Radiat Environ Biophys, 1995, 34, 21-7), if the author wants to show that the author's speculation is correct.

  1. Store et al ([38], Store et al, Radiat Res, 114, 331-353, 1988) used thousands of mice and revealed that the risks of leukemia, lung carcinoma, female breast carcinoma were compatible between mice and atomic bomb survivors, while the risks of liver tumors in mice were about twice as high risk as that of humans. Given that the atomic bomb survivors should be exposed to the fission neutron associated with atomic bomb explosion, the data obtained by the milestone study [38] should be considered for risks of high LET. Then, how can the author explain the differences of cancer risks between humans and mice. Does the author's explanation fit the model proposed by the author?

  1. The exception (female lung cancer) that did not fit the proposed model may be explained by the scarcity of heavy ion tumor does response data in rodents. In other words, how can the author convince the readers in different fields with a new risk model, which ignore the historically important data for humankind, with a little data with the exception? At minimum, the authors should tone down statements regarding to these points, otherwise, this study may mislead the direction with limited data.

  1. It is known that the cancer predisposition in human and mice are different (Parmar and Cunha, Endocr Relat Cancer, 2004, 11, 437-58). For example, BRCA-deficient mice did not show the increases in breast tumors (Evers and Jonkers, Oncogene, 2006, 25, 5885-97). Therefore, the basic information on the difference among species on cancer risks should be introduced and discussed.

  1. NTE might include a lot of factors such as bystander effects. How much is known about NTE in the space? The uncertainty of NTE in the space will make the new model difficult to predict. For example, the biological effects of the sun flare explosion in the space flight would be very huge and such an accidental factor should be considered if the author wants to establish the risk prediction models for astronauts.

Minor points:

  1. Some of the abbreviations should be explained.
  2. Epi: epidemiology studies in line 70.
  3. DERR (Direct excess relative risk) model in line 84.
  4. PDF (Probability distribution functions) estimates in line 245.

  1. The unit should be identified.
  2. a) 600 days in line 122 and 600 d in line 121
  3. b) 2.9% and 10 percent in line 122

  1. I recognize the difference among B6CF1 (Fiss. Neutrons) and the rest of data in Figure 1, panel D. Therefore, it is not convincing to describe “which show very similar response” in line 141.

  1. It is hard to understand what the parameters in Table 3 mean. What does it mean two Adjusted R2?

  1. Where can I find the description and insertion of Figure 4 in the manuscript?

  1. Although the tissue weighting factors for liver and lung are 0.04 and 0.12, respectively, why does REIC liver is much lower than REIC lung, and REIC breast in Table 5?

  1. I could not find the F:M ratio of 2.83 in [21]. I am afraid that the author wanted to cite Cahoon et al, Radiat Res, 2017, 187, 538-548.

  1. Why is the ERR of chronic exposure to fission neutrons higher than that of acute ERR?

  1. There are several mistakes.
  2. Spaces between assessment and Relative in line 40.
  3. (Fry, 1993, Cucinotta and Wilson, 1995) should be deleted.
  4. SD rat experiment is [19] not [8] in line 128.
  5. [6] in line 191 and [7] in line 195 should be replaced.
  6. “Carnes developed” should be “Carnes et al. developed” in 253.

Author Response

I thank the Reviewer for his or her critiques and suggestions. I have responded to each point as described below.

This paper tried to estimate risk prediction with a model of ERR based on the data obtained from animal studies with high LET radiation that avoids low LET (DERR model), which is uncertain at low dose range, to predict lifetime risks to astronauts. Interestingly, the data for male liver cancer and female breast cancer risk indicated the similar estimates between a QF model with NTE (NSCR-2020 with NTE) and DERR model. In contrast, the data for female lung cancer risk were 5-fold differences between DERR model and NSCR-2020 with NTE. The model fitting approach is interesting and might be relevant to exposures to high LET radiation in the space, but there are a number of major concerns about the interpretation of data that the authors need to address:

Specific comments are as follows.

Major points:

  1. Historically, a lot of studies with tremendous efforts have been done by scientists, particularly epidemiologists using the data of low LET based on atomic bomb survivors and nuclear accidents. Therefore, it is important to cite original important studies (e.g. Grant et al, Radiat Res, 2017, 187(5), 513-537) in parallel with the reports provided by the NCRP and BEIR to acknowledge their works.

 There were already a few of these publications cited([21, 43, 44], including paper by Grant et al. [43].

  1. For readers who are not familiar with the field of radiation effects, it is hard to find the sentence the author wants to indicate in the references because NCRP reports consist with hundreds of pages and is time consuming to search. Therefore, it would be helpful if the author provides the precise information on the sentence in the NCRP reports (e.g. NCRP Report 132, page xxx, line xxx). Moreover, the readers may wonder whether report published in 1989 [3], 1990 [5] is updated or not.

 I think it would be too many pages since most of the reports are about the experimental data and methods to make risk estimates.

  1. I understand that the biological effect data on the high LET in rodents is limited. However, the modelling attempt to fit the DERR model to the NSCR-2020 with NTE for assessing high LET is not sound to me, as one of the three cases (female lung cancer, male liver cancer, female breast cancer) failed to fit the model perfectly. Therefore, it would be interesting to examine another cancer case in rodents. I think this is an essential point, especially when the author wants to ignore the effect of low LET to substantiate the new risk model to assess the risk of high LET for space flight.

In Table 3, 8 of 9 data sets considered in the Saturation model fit fairly well, while one data set (liver cancer in B6CF1 mice with fission neutrons was poorly fit in the saturation model and somewhat better in the linear model. Here the data shows a lot of scatter at the lowest doses, and lack of data at a higher dose to establish a saturation is occurring.

The only other tumor where heavy ion (and fission neutrons) exist is AML. But here the effects are very similar to gamma-rays (Weil et al studies and older ones by Ullrich). I added discussion here.

  1. I am not convinced that the author discussed the reason for the difference between DERR model and NSCR-2020 with NTE in female lung cancer as the role of primary and secondary exposures to tabaco products. Globally, smoking rates among females are lower than those among males in human (Ng et al, JAMA, 2014, 311, 183-92). Therefore, it is important to check the risk of lung cancer in male mice using the data Dr. Fry provided who assessed the risks of cancers induced by neutron in thousands of male mice ([7] Storer and Fry, Radiat Environ Biophys, 1995, 34, 21-7), if the author wants to show that the author's speculation is correct.

I am a only pointing out the difference not declaring a solution to the difference. For male mice there are no heavy ion studies on lung cancer risk. The LSS and Mayak studies show a higher radiation risk for female lung cancer compared to males.

  1. Store et al ([38], Store et al, Radiat Res, 114, 331-353, 1988) used thousands of mice and revealed that the risks of leukemia, lung carcinoma, female breast carcinoma were compatible between mice and atomic bomb survivors, while the risks of liver tumors in mice were about twice as high risk as that of humans. Given that the atomic bomb survivors should be exposed to the fission neutron associated with atomic bomb explosion, the data obtained by the milestone study [38] should be considered for risks of high LET. Then, how can the author explain the differences of cancer risks between humans and mice. Does the author's explanation fit the model proposed by the author?

 It is well documented in the literature that fission neutrons are very small component of the doses in the LSS study, and the influence of the much higher gamma-ray dose dominates. Also, heavy ions impart energy in tissues over long tracks while the high LET protons from neutron exposures are short tracks in tissues.

  1. The exception (female lung cancer) that did not fit the proposed model may be explained by the scarcity of heavy ion tumor does response data in rodents. In other words, how can the author convince the readers in different fields with a new risk model, which ignore the historically important data for humankind, with a little data with the exception? At minimum, the authors should tone down statements regarding to these points, otherwise, this study may mislead the direction with limited data.

 We re-worded the discussion in the revised manuscript. The low LET studies were noted as valuable in the introductions. Also one of the major points of the paper is the QF and not just epidemiology data. The arguments to look at a different approach are already mentioned in the Introduction and Discussion sections:

  1. No human data for heavy ions and very little for low energy high LET radiation.
  2. In Medical science an unknown risk is studied in experimental models using some form a Relative risk model. I use a retrospective approach but certainty a prospective design would be better.
  3. Experimental models show many qualitative differences between high and low LET radiation, so using low LET epidemiology, and QF is suspect since it ignores the qualitative agreement altogether.

In addition, the low LET epidemiology does not have to be ignored and can always be combined in a different approach with a RR model, perhaps a hybrid or Bayesian approach. Some discussion added here.  

  1. It is known that the cancer predisposition in human and mice are different (Parmar and Cunha, Endocr Relat Cancer, 2004, 11, 437-58). For example, BRCA-deficient mice did not show the increases in breast tumors (Evers and Jonkers, Oncogene, 2006, 25, 5885-97). Therefore, the basic information on the difference among species on cancer risks should be introduced and discussed.

 Many review articles suggest there are many genetic overlaps between humans and mice as well. I had already discussed briefly and cited [44] using collaborative cross mouse colonies. Added other discussion in the revision. None of the papers mentioned by Reviewer are for high LET or heavy ions or low dose or low dose-rates so meaning is unclear. Also, there are many other studies by for eg. Peter Demant that suggest extensive genetic overlap between mice and humans. In any case I added references here.

  1. NTE might include a lot of factors such as bystander effects. How much is known about NTE in the space? The uncertainty of NTE in the space will make the new model difficult to predict. For example, the biological effects of the sun flare explosion in the space flight would be very huge and such an accidental factor should be considered if the author wants to establish the risk prediction models for astronauts.

Solar flares protons, which are largely low LET radiation, are shielded easily because of the lower energies, which has been reported already. Here most of the solar flare flux is below 100 MeV and spacecraft sufficient shielding. This is mostly an operational issue using real-time dosimetry to ensure an adequate response. These topics are not in scope of this manuscript.  NTE implies several area including bystander effects and tissue microenvironment changes.

Minor points:

 The following were corrected in the revised paper as suggested:

  1. Some of the abbreviations should be explained.
  2. Epi: epidemiology studies in line 70.
  3. DERR (Direct excess relative risk) model in line 84.
  4. PDF (Probability distribution functions) estimates in line 245.

PDF was defined on line 104.

  1. The unit should be identified.
  2. a) 600 days in line 122 and 600 d in line 121
  3. b) 2.9% and 10 percent in line 122

 Corrected.

  1. I recognize the difference among B6CF1 (Fiss. Neutrons) and the rest of data in Figure 1, panel D. Therefore, it is not convincing to describe “which show very similar response” in line 141.

 Ok modified.

  1. It is hard to understand what the parameters in Table 3 mean. What does it mean two Adjusted R2?

The two values are for the two models that are compared. I added a new sub-heading in the table to avoid confusion.

  1. Where can I find the description and insertion of Figure 4 in the manuscript?

 Fixed (the IJMS template cause an issue on missing short paragraph on Fig 4)

  1. Although the tissue weighting factors for liver and lung are 0.04 and 0.12, respectively, why does REIC liver is much lower than REIC lung, and REIC breast in Table 5?

 This is due to differences in background cancer rates for lung and liver cancer.

  1. I could not find the F:M ratio of 2.83 in [21]. I am afraid that the author wanted to cite Cahoon et al, Radiat Res, 2017, 187, 538-548.

 Corrected (ref 21 and ref 44 where flipped).

  1. Why is the ERR of chronic exposure to fission neutrons higher than that of acute ERR?

 This has been described as NTE or inverse dose-rate effect in the literature.

We have corrected the following in the revised text -

  1. There are several mistakes.
  2. Spaces between assessment and Relative in line 40.
  3. (Fry, 1993, Cucinotta and Wilson, 1995) should be deleted.
  4. SD rat experiment is [19] not [8] in line 128.
  5. [6] in line 191 and [7] in line 195 should be replaced.
  6. “Carnes developed” should be “Carnes et al. developed” in 253.

Reviewer 2 Report

This manuscirpt seems not focused on any molecular events based on the Materrials and Methods, as well as the Results description. It may be interested in people who are working in radiation protection or radiation physics, but not molecular sciences. Therefore, I do not recommend this manuscript to be submitted to IJMS but other specific journals. 

Another suggestion is this manuscript is a single author's works, so no need to use "we".

Author Response

This manuscirpt seems not focused on any molecular events based on the Materrials and Methods, as well as the Results description. It may be interested in people who are working in radiation protection or radiation physics, but not molecular sciences. Therefore, I do not recommend this manuscript to be submitted to IJMS but other specific journals. 

Another suggestion is this manuscript is a single author's works, so no need to use "we".

I thank the Reviewer for their comments.

The focus of the present report is the major goal and application of molecular studies on cancer risks from heavy ion and other high LET radiation. We have added discussion in the revised report in this area.

I have modified use of “we” as suggested

Reviewer 3 Report

This manuscript advocates an excess relative risk (ERR) model to estimate and predict cancer risk for space travel not based on extrapolation on low-LET radiation data but rather on animal data from high-LET exposure.  Current models rely on quality factors and adjustements for dose rate which entail significant uncertainties.  More importantly, they do not take into account the peculiar radiobiological response to high-LET radiation, nor do they account for non-targeted effects (NTEs). The author is aware that limitations using such an approach do exist but also suggests possible improvements, such as developments of specific animal model encompassing the most significant radiation-induced tumours in humans as well as  using more detailed models of neutron transport.

Only a very minor number of suggestions:

  1. The author seems to dislike the use of hyphenation: there are countless cases where I would like to see something like "radiation-induced" rather than "radiation induced" and similar; yet, not being a native English speaker, I just feel I had to point this out.
  2. There are no keywords, at least in the mansucript version I received for peer review
  3.  Lines 38-39, Introduction: please consider replace are with being in  sentence "with missions outside the protection of the Earth’s magnetic field are the main challenge for risk assessment"
  4. Line 44: Consider commas at the beginning and the ned of sentence "most frequently the Lifespan Study (LSS) of atomic-bomb survivors"
  5. 5. Line 72: consider putting E as subscript in aE following formula (1)
  6. Lines 90-91 Isn't the following sentence missing something "The model is de-coupled from radiation epidemiology data although observations."?
  7.  Line 139: Please use third person in "the lean model fit" i.s. fits
  8. Table 2 Please correct typo "heave" instead of "heavy"
  9. Line 329 please correct typo "reports" in "report"

Author Response

I thank the reviewer for their comments. I have responded in total agreement with their suggestions as outlined below.

This manuscript advocates an excess relative risk (ERR) model to estimate and predict cancer risk for space travel not based on extrapolation on low-LET radiation data but rather on animal data from high-LET exposure.  Current models rely on quality factors and adjustments for dose rate which entail significant uncertainties.  More importantly, they do not take into account the peculiar radiobiological response to high-LET radiation, nor do they account for non-targeted effects (NTEs). The author is aware that limitations using such an approach do exist but also suggests possible improvements, such as developments of specific animal model encompassing the most significant radiation-induced tumours in humans as well as  using more detailed models of neutron transport.

Only a very minor number of suggestions:

  1. The author seems to dislike the use of hyphenation: there are countless cases where I would like to see something like "radiation-induced" rather than "radiation induced" and similar; yet, not being a native English speaker, I just feel I had to point this out.

Yes made such changes.

  1. There are no keywords, at least in the mansucript version I received for peer review

Yes, an oversight in submission and keywords now added.

  1.  Lines 38-39, Introduction: please consider replace are with being in  sentence "with missions outside the protection of the Earth’s magnetic field are the main challenge for risk assessment"

Yes corrected.

  1. Line 44: Consider commas at the beginning and the need of sentence "most frequently the Lifespan Study (LSS) of atomic-bomb survivors"

Yes

  1. 5. Line 72: consider putting E as subscript in aE following formula (1)

There are two variables attained age, a, and age at exposure aE.

  1. Lines 90-91 Isn't the following sentence missing something "The model is de-coupled from radiation epidemiology data although observations."?

Yes a typo now corrected.

  1.  Line 139: Please use third person in "the lean model fit" i.s. fits

Yes corrected.

  1. Table 2 Please correct typo "heave" instead of "heavy"

Yes corrected.

  1. Line 329 please correct typo "reports" in "report"

Yes corrected.

Round 2

Reviewer 1 Report

I thank the Reviewer for his or her critiques and suggestions. I have responded to each point as described below.

This revision mainly improves the introduction, results and discussion. Although, the author addressed some concerns, the major and key concerns have not been addressed, which undermines confidence in the proposed model. If key concerns are not addressed faithfully, this study cannot be accepted in terms of the credibility of the model.

This paper tried to estimate risk prediction with a model of ERR based on the data obtained from animal studies with high LET radiation that avoids low LET (DERR model), which is uncertain at low dose range, to predict lifetime risks to astronauts. Interestingly, the data for male liver cancer and female breast cancer risk indicated the similar estimates between a QF model with NTE (NSCR-2020 with NTE) and DERR model. In contrast, the data for female lung cancer risk were 5-fold differences between DERR model and NSCR-2020 with NTE. The model fitting approach is interesting and might be relevant to exposures to high LET radiation in the space, but there are a number of major concerns about the interpretation of data that the authors need to address:

Specific comments are as follows.

Major points:

  1. Historically, a lot of studies with tremendous efforts have been done by scientists, particularly epidemiologists using the data of low LET based on atomic bomb survivors and nuclear accidents. Therefore, it is important to cite original important studies (e.g. Grant et al, Radiat Res, 2017, 187(5), 513-537) in parallel with the reports provided by the NCRP and BEIR to acknowledge their works.

 There were already a few of these publications cited([21, 43, 44], including paper by Grant et al. [43].

For readers who are not familiar with the field of radiation effects, it is hard to find the sentence the author wants to indicate in the references because NCRP reports consist with hundreds of pages and is time consuming to search. Therefore, it would be helpful if the author provides the precise information on the sentence in the NCRP reports (e.g. NCRP Report 132, page xxx, line xxx). Moreover, the readers may wonder whether report published in 1989 [3], 1990 [5] is updated or not.

 I think it would be too many pages since most of the reports are about the experimental data and methods to make risk estimates.

The author did not answer to the question. “Moreover, the readers may wonder whether report published in 1989 [3], 1990 [5] is updated or not.”

  1. I understand that the biological effect data on the high LET in rodents is limited. However, the modelling attempt to fit the DERR model to the NSCR-2020 with NTE for assessing high LET is not sound to me, as one of the three cases (female lung cancer, male liver cancer, female breast cancer) failed to fit the model perfectly. Therefore, it would be interesting to examine another cancer case in rodents. I think this is an essential point, especially when the author wants to ignore the effect of low LET to substantiate the new risk model to assess the risk of high LET for space flight.

In Table 3, 8 of 9 data sets considered in the Saturation model fit fairly well, while one data set (liver cancer in B6CF1 mice with fission neutrons was poorly fit in the saturation model and somewhat better in the linear model. Here the data shows a lot of scatter at the lowest doses, and lack of data at a higher dose to establish a saturation is occurring.

The only other tumor where heavy ion (and fission neutrons) exist is AML. But here the effects are very similar to gamma-rays (Weil et al studies and older ones by Ullrich). I added discussion here.

I do not agree with the author’s comment because lung cancer in B6CF1 female mice with fission neutrons shows less fit in the saturation model (p=0.585) and better in the linear model (p=0.792). Moreover, the author concluded that the Saturation model fit fairly well in female Balb/c mice exposed to fission neutrons (FN), which shows a scatter at the lower doses, and lack of data at a higher dose to establish a saturation is occurring. Such an inconsistency undermines confidence in the proposed model.

In addition, a considerable number of heavy ion and fission neutron-induced mouse cancer research papers have been published as follows

  1. Lymphoreticular tumors, epithelial tissue tumors, lung tumors, B6CF1 female, Radiat Res, 104, 420-428, 1985
  2. Thymic and nonthymic lymphoma, lung tumors, B6CF1 male, Radiat Res, 103, 77-88, 1985
  3. Lymphoreticular tissue tumors, vascular tissue tumors, epithelial tissue tumors, lung tumors, liver tumors, ovarian tumors in B6CF1 male & female, Radiat Res, 129, 19-36, 1992
  4. Thymoma, leukemia, carcinoma, sarcoma in C57BL male, Radiat Res, 113, 300-317, 1988
  5. Intestinal carcinoma in LAF1 male & female, Cancer Res, 16, 873–876, 1956
  6. Leukemia, lymphoma, ovarian tumor, pulmonary tumor in RF/Un male & female, Radiat Res, 41, 467-491, 1970
  7. Leukemia, ovarian tumor, lung tumor in RF male & female, Cancer Res, 14, 682-90, 1954
  8. Leukemia, lymphoma in B6CF1 male, Radiat, Res, 119, 553-561, 1989
  9. Harderian gland tumor, lymphoma in CBA/JCR HSD male, Radiat Res, 169, 615–625, 2008
  10. Lung tumor in C57BL/6 male & female, Radiat Res, 183, 233-9, 2015
  11. Leukemia, epithelial tumor in BC3F1 male, Radiat Res, 138, 252-259, 1994
  12. Gastrointestinal tumor in (C57LXA)F1 male & female, Radiat Res, 11, 545-556, 1959.

There should be some studies that I haven't been introduced yet, because I can easily search and find so many studies. At least, the data of these studies should be used to evaluate the proposed model is convincing. If these data are not used, appropriate reasons should be given.

  1. I am not convinced that the author discussed the reason for the difference between DERR model and NSCR-2020 with NTE in female lung cancer as the role of primary and secondary exposures to tabaco products. Globally, smoking rates among females are lower than those among males in human (Ng et al, JAMA, 2014, 311, 183-92). Therefore, it is important to check the risk of lung cancer in male mice using the data Dr. Fry provided who assessed the risks of cancers induced by neutron in thousands of male mice ([7] Storer and Fry, Radiat Environ Biophys, 1995, 34, 21-7), if the author wants to show that the author's speculation is correct.

I am a only pointing out the difference not declaring a solution to the difference. For male mice there are no heavy ion studies on lung cancer risk. The LSS and Mayak studies show a higher radiation risk for female lung cancer compared to males.

An unpersuasive speculation should be omitted. At least, it is important to compare the data on the risk of lung cancer in thousands of male mice and thousands female mice using the data Dr. Fry provided ([7] Storer and Fry, Radiat Environ Biophys, 1995, 34, 21-7) for assessing the new model.

In the LSS studies, the ERR for males was significantly lower than that for females due to lower background rates of cancer incidence (not just lung cancer) among females (Radiat Res, 2017, 187, 513-537). In the Mayak studies, considerable statistical uncertainty exists in risk estimates for females (Radiat Res, 2013, 179, 332-42; Radiat Res, 195, 334–346, 2021). The lower baseline risk and the higher levels of plutonium measured in females makes higher solid cancer (liver cancer, lung cancer, bone cancer) risks for females than for males (Radiat Res, 149, 366-371, 1998; Radiat Res, 154, 246–252, 2000; Radiat Res, 154, 246–252, 2000). Moreover, the ERR for lung cancer was 0.16 (0.01, 0.32) among the 55,218 male workers and 0.09 (-0.19, 0.36) among the 53,801 female workers in the Medical Radiation Workers in the United States, 1965-2016 (Int J Radiat Biol, 2021, 1-63). Taken together, it is not scientifically accepted that a higher radiation risk for female lung cancer compared to males.

  1. Store et al ([38], Store et al, Radiat Res, 114, 331-353, 1988) used thousands of mice and revealed that the risks of leukemia, lung carcinoma, female breast carcinoma were compatible between mice and atomic bomb survivors, while the risks of liver tumors in mice were about twice as high risk as that of humans. Given that the atomic bomb survivors should be exposed to the fission neutron associated with atomic bomb explosion, the data obtained by the milestone study [38] should be considered for risks of high LET. Then, how can the author explain the differences of cancer risks between humans and mice. Does the author's explanation fit the model proposed by the author?

 It is well documented in the literature that fission neutrons are very small component of the doses in the LSS study, and the influence of the much higher gamma-ray dose dominates. Also, heavy ions impart energy in tissues over long tracks while the high LET protons from neutron exposures are short tracks in tissues.

  1. The exception (female lung cancer) that did not fit the proposed model may be explained by the scarcity of heavy ion tumor does response data in rodents. In other words, how can the author convince the readers in different fields with a new risk model, which ignore the historically important data for humankind, with a little data with the exception? At minimum, the authors should tone down statements regarding to these points, otherwise, this study may mislead the direction with limited data.

 We re-worded the discussion in the revised manuscript. The low LET studies were noted as valuable in the introductions. Also one of the major points of the paper is the QF and not just epidemiology data. The arguments to look at a different approach are already mentioned in the Introduction and Discussion sections:

  1. No human data for heavy ions and very little for low energy high LET radiation.
  2. In Medical science an unknown risk is studied in experimental models using some form a Relative risk model. I use a retrospective approach but certainty a prospective design would be better.
  3. Experimental models show many qualitative differences between high and low LET radiation, so using low LET epidemiology, and QF is suspect since it ignores the qualitative agreement altogether.

In addition, the low LET epidemiology does not have to be ignored and can always be combined in a different approach with a RR model, perhaps a hybrid or Bayesian approach. Some discussion added here.  

  1. It is known that the cancer predisposition in human and mice are different (Parmar and Cunha, Endocr Relat Cancer, 2004, 11, 437-58). For example, BRCA-deficient mice did not show the increases in breast tumors (Evers and Jonkers, Oncogene, 2006, 25, 5885-97). Therefore, the basic information on the difference among species on cancer risks should be introduced and discussed.

 Many review articles suggest there are many genetic overlaps between humans and mice as well. I had already discussed briefly and cited [44] using collaborative cross mouse colonies. Added other discussion in the revision. None of the papers mentioned by Reviewer are for high LET or heavy ions or low dose or low dose-rates so meaning is unclear. Also, there are many other studies by for eg. Peter Demant that suggest extensive genetic overlap between mice and humans. In any case I added references here.

  1. NTE might include a lot of factors such as bystander effects. How much is known about NTE in the space? The uncertainty of NTE in the space will make the new model difficult to predict. For example, the biological effects of the sun flare explosion in the space flight would be very huge and such an accidental factor should be considered if the author wants to establish the risk prediction models for astronauts.

Solar flares protons, which are largely low LET radiation, are shielded easily because of the lower energies, which has been reported already. Here most of the solar flare flux is below 100 MeV and spacecraft sufficient shielding. This is mostly an operational issue using real-time dosimetry to ensure an adequate response. These topics are not in scope of this manuscript.  NTE implies several area including bystander effects and tissue microenvironment changes.

Minor points:

 The following were corrected in the revised paper as suggested:

  1. Some of the abbreviations should be explained.
  2. Epi: epidemiology studies in line 70.
  3. DERR (Direct excess relative risk) model in line 84.
  4. PDF (Probability distribution functions) estimates in line 245.

PDF was defined on line 104.

  1. The unit should be identified.
  2. a) 600 days in line 122 and 600 d in line 121
  3. b) 2.9% and 10 percent in line 122

 Corrected.

  1. I recognize the difference among B6CF1 (Fiss. Neutrons) and the rest of data in Figure 1, panel D. Therefore, it is not convincing to describe “which show very similar response” in line 141.

 Ok modified.

  1. It is hard to understand what the parameters in Table 3 mean. What does it mean two Adjusted R2?

The two values are for the two models that are compared. I added a new sub-heading in the table to avoid confusion.

  1. Where can I find the description and insertion of Figure 4 in the manuscript?

 Fixed (the IJMS template cause an issue on missing short paragraph on Fig 4)

  1. Although the tissue weighting factors for liver and lung are 0.04 and 0.12, respectively, why does REIC liver is much lower than REIC lung, and REIC breast in Table 5?

 This is due to differences in background cancer rates for lung and liver cancer.

  1. I could not find the F:M ratio of 2.83 in [21]. I am afraid that the author wanted to cite Cahoon et al, Radiat Res, 2017, 187, 538-548.

 Corrected (ref 21 and ref 44 where flipped).

As I mentioned above, the ERR for males was significantly lower than that for females due to lower background rates of cancer incidence (not just lung cancer) among females (Radiat Res, 2017, 187, 513-537) in the LSS studies. Therefore, these points should be described in the manuscript.

  1. Why is the ERR of chronic exposure to fission neutrons higher than that of acute ERR?

 This has been described as NTE or inverse dose-rate effect in the literature.

To make the manuscript better for the reader, it is important to introduce the reverse dose-rate effect (IDRE) and cite the literature (eg. Hill et al, Nature, 298, 67–69, 1982) describing the IDRE to explain the reason why.

We have corrected the following in the revised text -

  1. There are several mistakes.
  2. Spaces between assessment and Relative in line 40.
  3. (Fry, 1993, Cucinotta and Wilson, 1995) should be deleted.
  4. SD rat experiment is [19] not [8] in line 128.

Although, you mentioned that you have corrected, this has not been corrected yet. SD rat experiment is [19] not [5] in line 133. Be responsible, sincere and attentive to your manuscript.

  1. [6] in line 191 and [7] in line 195 should be replaced.
  2. “Carnes developed” should be “Carnes et al. developed” in 253.

Author Response

Responses in the attached file.

Reviewer 2 Report

Although the author explained that "The focus of the present report is the major goal and application of molecular studies on cancer risks from heavy ion and other high LET radiation.", this report may be used for prediction of cancer risk but not molecular events. Addition of statements does not solve the original problems.  It is also confused if this study is to predict the cancer risk of Space travel or to understand the high LET based cancer therapy? In conclusion, it seems no any perspective application of this model to molecular study. I still believe this report is better fit to a more specific type of journal. 

Author Response

Thanks for your comments. The Editors have agreed to review the manuscript for this issue. 

Round 3

Reviewer 1 Report

This time, the author addressed the major concerns and the revised manuscript has been improved. As the author admitted, there are many uncertainties in the analysis with the limited data available so far, and the authors' claims should be generally toned down before publication.